# Advanced Electrospinning Technology Applied to Polymer-Based Sensors in Energy and Environmental Applications

**DOI:** 10.3390/polym16060839

**Published:** 2024-03-19

**Authors:** Gang Lu, Tao Tian, Yuting Wang

**Affiliations:** 1School of Energy and Environmental Engineering, University of Science and Technology Beijing, Beijing 100083, China; bkhjlg@163.com; 2School of Precision Instrument and Opto-Electronics Engineering, Tianjin University, Tianjin 300072, China; taotian@tju.edu.cn; 3School of Mathematics and Physics, University of Science and Technology Beijing, Beijing 100083, China

**Keywords:** electrospinning, polymer, sensors, energy collection, environmental applications

## Abstract

Due to its designable nanostructure and simple and inexpensive preparation process, electrospun nanofibers have important applications in energy collection, wearable sports health detection, environmental pollutant detection, pollutant filtration and degradation, and other fields. In recent years, a series of polymer-based fiber materials have been prepared using this method, and detailed research and discussion have been conducted on the material structure and performance factors. This article summarizes the effects of preparation parameters, environmental factors, a combination of other methods, and surface modification of electrospinning on the properties of composite nanofibers. Meanwhile, the effects of different collection devices and electrospinning preparation parameters on material properties were compared. Subsequently, it summarized the material structure design and specific applications in wearable device power supply, energy collection, environmental pollutant sensing, air quality detection, air pollution particle filtration, and environmental pollutant degradation. We aim to review the latest developments in electrospinning applications to inspire new energy collection, detection, and pollutant treatment equipment, and achieve the commercial promotion of polymer fibers in the fields of energy and environment. Finally, we have identified some unresolved issues in the detection and treatment of environmental issues with electrospun polymer fibers and proposed some suggestions and new ideas for these issues.

## 1. Introduction

The energy crisis and environmental issues are some of the most serious and important global issues facing humanity [1]. With the rapid development of industrialization, urbanization, and modern agriculture, the damage caused by waste and intermediate products in the production process to the ecosystem is collectively referred to as environmental pollution. With the increasing environmental pollution, new products produced by humans are constantly accompanied by the generation of new pollutants, which have far exceeded the self-cleaning ability of the environment. Among them, water and air pollution are the most prominent problems among the two systems in the environment. Due to water being one of the essential elements for human life and development, and being a limited and easily polluted resource, water pollution and reduced freshwater supply are becoming key global issues [2]. Similarly, the impact of air quality on the human living environment is particularly important. Air pollution mainly includes pollution caused by small particulate matter, nitrogen dioxide, sulfur dioxide, carbon monoxide, and other pollutants in the air that affect the environment [3,4]. With the increasing amount of air pollution, the World Health Organization points out that the number of diseases caused by air pollution is constantly increasing every year [5]. Therefore, accurate real-time measurement and effective governance of the atmospheric air environment are urgent.

Polymer-based devices manufactured through electrospinning have developed into a complex set of technologies since their inception. Advanced electrospinning technologies have received widespread attention due to their potential for energy and environmental applications. Many flexible and wearable electronic devices have been used to detect health-related signals, such as pressure/strain sensors, temperature sensors, humidity sensors, gas sensors, integrated sensing platforms, etc. [6,7].

Among them, piezoelectric sensing systems can detect human motion and small vibrations while collecting and storing energy through piezoelectric energy conversion [8,9]. For environmental detection and governance, various real-time detection sensing devices and pollutant degradation absorption materials have been developed in sequence [10,11].

In recent years, polymer-based electrospun nanomaterials have developed rapidly in the fields of energy collection and environmental purification. So far, a series of polymer nanomaterials with different structures and functions, such as nanowires, nanotubes, nanorods, and nanobelts, have been prepared, and their potential applications in energy and environmental crisis-related fields have been reported, including gas detection [12], humidity detection [13,14], pollutant detection [15], air purification [16], water purification [17], and other related research [18]. There are many preparation methods for the production of nanomaterials, among which electrospinning technology is a very promising approach [19]. Electrospinning, as a convenient, fast, and environmentally friendly preparation method, has a wide range of applications in the design and manufacturing of new materials for human motion monitoring and energy harvesting, environmental monitoring, pollution purification, and other fields [20]. The fibers prepared by this method have obvious advantages such as high specific surface area, low specific gravity, high porosity, and designable nanostructures and surface morphology, making it a highly commercial preparation method. By adjusting the electrospinning process parameters and modifying the collection device, various synthetic polymers, composite materials, ceramic materials, as well as various polymer-based functional materials ranging from micrometers to nanometers can be obtained [21,22]. With the increasing demand for energy shortages and environmental restoration, it is necessary to conduct statistics and summarize the current application status and shortcomings of this method in the field of environment, providing inspiration and guidance for future development directions. Therefore, the focus of this review is to provide an overview of the latest research on the effects of different structural designs and preparation parameters on the performance of electrospun polymer nanofibers, as well as their applications in energy and the environment.

## 2. Advanced Electrospinning of Polymer-Based Nanofibers

Electrospinning utilizes high-voltage electrostatic field forces to form Taylor cones from polymer solutions or melts, which are then stretched into countless nanoscale ultrafine fibers, and then cured through solvent evaporation or melt solidification [23,24,25]. By electrospinning, not only can a morphology-controllable nanofiber membrane be obtained, but the membrane also has a relatively large specific surface area and high porosity. Recently, many studies have been published on the use of electrospinning to manufacture piezoresistive nanofiber sensors. Further improvement of polymer functional nanofibers based on electrospinning can be achieved through surface modification and subsequent conversion of nanofibers obtained through electrospinning into derivatives. We will provide a detailed introduction and discussion of the latest surface modification methods and structures in Section 2.1. The preparation of nanofiber derivatives is an additional process, as described in Section 2.1.

### 2.1. Structural Design

At present, nanomaterials prepared using electrospinning technology have advantages such as low cost, multiple material options, controllable parameters, and large-scale production, making it one of the most promising manufacturing technologies for commercial preparation of controllable nanostructured materials in the future [26]. The fibers prepared by this method also have characteristics such as a large specific surface area, high porosity, excellent mechanical properties, designable structure, and optimized performance. In recent years, further improvements have been made to high-pressure nozzles and collection devices, resulting in nanomaterials with various structures, such as rods, belts, hollow, core-shell, porous, spiral, sponges, etc. [27].

Nayak et al. utilized [28] high-temperature electrospinning and post-treatment methods by a rotary rod collector to prepare axially elongated microfibrillar yarn. It is found that post-processing reduces fiber diameter while significantly improving molecular chain orientation, forming a microfiber structure that extends in the stretching direction. The increased crystallinity and molecular orientation from uniaxial stretching contribute to improving the mechanical properties of fiber materials. At the same time, the molecular entanglement of the fibers is eliminated during the melting and recrystallization processes, forming oriented thin sheets, and significantly improving the tensile strength and Young’s modulus of the fibers.

Ribbon-structured fibers are mainly obtained by adjusting the precursor solution and the electrospinning process. If a multi-layer structure is required, further treatment can be carried out. Li et al. [29] added tetraethyl orthosilicate (TEOS) to the precursor solution of electrospinning, allowing for the formation of an insensitive network structure in the original nanoribbons during the electrospinning process, where the TEOS comes into contact with water in the air through solvent evaporation. In the subsequent fluorination process, as the calcination temperature slowly increases, the solvent continues to evaporate. TEOS undergoes hydrolysis condensation and decomposition with H_2_O and O_2_ in the air at high temperatures, producing SiO_2_ on the surface of the nanobelt. Anions remain inside the nanobelt, gradually forming a dense SiO_2_ network structure on the surface of the nanobelt, ultimately forming a SiO_2_ shell layer.

Coaxial electrospinning is an important improvement method in electrospinning, which uses a core-shell spinneret to spray different precursor solutions from the channel [30,31]. By further processing to remove or retain the inner solution, nanofibers with core shells, pores, or hollow structures can be obtained [32,33]. Shao et al. [34] prepared PVDF hollow fibers using coaxial electrospinning and quantified the wall thickness of the hollow fibers by adjusting the internal solution concentration. Meanwhile, with the design and control of different structures, their electrical performance is regulated. Scaffaro et al. [35] prepared membrane structures with controllable structure and porosity using coaxial wet electrospinning with in-situ leaching. Hollow and porous fibers can be obtained through further processing. At the same time, the influence of porous structure on the mechanical properties, wettability, oil absorption capacity, and reusability of the membrane was evaluated.

### 2.2. Preparation of the Derivatives

Compared with directly prepared electrospun nanomaterials, electrospun derivatives obtained through the addition or post-modification of other materials exhibit certain specific advantages, such as high yield, excellent performance, special functions, multifunctionality, and a high surface volume ratio, thus attracting the attention and research of researchers on these materials [36]. The continuous emergence and use of new functional materials, such as metal-organic frameworks (MOFs), graphene, carbon nanotubes, metal nanofibers, etc., provide more possibilities for their application in the fields of environment and energy.

MOFs have significant potential applications in wastewater treatment and pollutant adsorption due to their high specific surface area, diverse structures and functions, diverse material choices, and unique characteristics [37,38]. However, the nanoscale particle size and fine powder form of MOFs are difficult to recover, resulting in secondary pollution of the environment. The nanofibers prepared by electrospinning have the characteristics of formability, spinnability into woven fabrics, and large-area preparation. Therefore, the combination of electrospinning materials and MOFs to prepare high-performance, environmentally friendly derivative materials has sparked a research boom [23]. Zhang et al. [39] proposed a macro structure assembly strategy, first preparing zeolite imidazole salt framework (ZIF-8)/nanofibers using the ZIF-8 nanoparticle method, and then incorporating it into the nanofibers through electrospinning. The obtained derivatives present zero-dimensional hollow structures, one-dimensional nanofibers, and three-dimensional carbon aerogels, simultaneously exhibiting low density, high mechanical strength, excellent adsorption performance, and excellent photothermal conversion ability. This provides an effective assembly method for constructing centimeter macroscopic structural components from nanomaterials in the future and is effectively applied in environmental protection and sewage purification.

Another derivative uses graphene as the basic material for composite improvement, and graphene has become a research hotspot due to its excellent electrical and mechanical properties. Due to the progress and breakthroughs of graphene in flexible substrate materials and conductive materials, graphene flexible pressure sensors have the advantages of lighter weight, more convenient use, higher sensitivity, and better stability. The substrate material exists as a support for sensors, and the excellent physical properties and lattice structure of graphene give it high electron mobility and good stretchability [40,41] Othman et al. [42] prepared polyacrylonitrile based activated carbon nanofibers by electrospinning polyacrylonitrile with different concentrations of reduced graphene oxide. The addition of rGO increases the specific surface area, total pore volume, and micropore volume of the fibers, thereby significantly enhancing the adsorption capacity of the composite material for carbon dioxide.

Carbon nanotubes are currently one of the most widely used flexible sensor materials due to their excellent conductivity, high strength, and flexible plasticity. Carbon nanotubes can be prepared into thin films by wet chemistry or drying methods, which can be used as sensitive layers in sensors and have good response sensitivity and stability [43]. Sriwichai et al. [44] prepared electrospun fiber membranes of poly (3-aminobenzylamine) (PABA)/functionalized multi-walled carbon nanotubes (f-CNTs) composites on screen printed electrodes for use as electrochemical glucose biosensors. The addition of f-CNTs significantly increases the electrochemical activity of the material, resulting in the obtained electrochemical glucose biosensor exhibiting good response sensitivity and high selectivity.

Hydrogel is a three-dimensional polymer network, which can expand rapidly in water. Hydrogels contain polymer networks similar to solids, and their aqueous phase contributes to the rapid diffusion of the carrier, showing liquid-like transport characteristics. In addition, the introduction of functional micro/nanostructures has significantly improved the key performance parameters of hydrogel-based devices. By combining the characteristics of hydrogel materials with nanofiber materials, electrospinning is used to design and prepare devices with unique properties and functions. Chen et al. [45] used the combination of electrospinning and 3D printing to draw the patterned borate on the electrospun film with mesoscopic structure to obtain the deformed hydrogel material with rapid deformation and enhanced 3D shape. This design utilizes the in-plane and interlayer stresses caused by expansion/contraction mismatch to achieve the deformation behavior of electrospun films while adapting to environmental changes. A series of fast deformation hydrogel actuators with various unique response behaviors have been constructed, such as reversible/irreversible deformation with 3D structure, 3D folding, and braking devices with multi-low-energy 3D structures.

## 3. Applications in Energy and Environmental Applications

Electrospun nanofibers are excellent materials with high specific surface area, flexibility, biocompatibility, and non-toxicity. The adjustable microstructures, such as three-dimensional structures, core-shell, hollow, porous, and nanonets, are being commercialized through simple methods for obtaining these structures and their continuous production characteristics. The production process of these fibers does not produce any excess waste products. Due to their high specific surface area and controllable structure, they have also attracted attention as sensing materials in environmental safety, energy collection, health detection, etc. The data in Figure 1 shows that the application of electrospinning in the fields of energy and environment has been increasing rapidly year by year, and has been growing rapidly since 2020. Therefore, this section summarizes the latest progress in various applications of electrospinning in these fields.

### 3.1. Human Motion Monitoring and Energy Harvesting

Over the past two decades, Electrospun nanofibers have had significant applications and extensive research in the field of wearable devices for monitoring health and activity [46]. Wearable electronic devices generally refer to flexible integrated electronic devices and wearable electronic devices on the body [47,48]. Due to its miniaturization, stretchability, comfortable wearing, and low power consumption, it has shown excellent application prospects for in-home or elderly care, patient physiological signal detection, and daily exercise body data detection for athletes and ordinary people [49,50]. In this regard, researchers have developed flexible integrated systems with multiple functions, such as smartwatches, smart clothing, blood sugar detection, heart rate detection, etc. However, currently, these devices typically require traditional battery power or regular charging, and the potential environmental pollution issues associated with the batteries they are applied to greatly limit their widespread application. Therefore, a series of nanofiber-based materials that can obtain energy from environmental sources (such as local body movements, solar energy, thermal energy, etc.) to provide energy for portable and wearable systems have emerged [51,52,53].

At present, various energy harvesting devices are used for body signal detection to collect mechanical energy, solar energy, and thermal energy (caused by temperature differences) during the process of life movement, including energy harvesting devices integrated based on principles such as piezoelectric, frictional, electromagnetic, solar cells, thermoelectric, etc. [54]. Among them, piezoelectric sensors and triboelectric sensors have attracted much attention due to their excellent characteristics and huge application prospects [55,56]. Due to the different mechanisms of the two types of sensing, the design of sensors prepared using the two mechanisms is also different. Triboelectric sensors usually require the design of two surfaces with different charge affinities to assist in contact and charge hole separation processes. The detection capabilities of the two sensors are similar in very high frequency and a wide frequency range.

In different materials, piezoelectric materials convert dynamic mechanical energy with high power density and durability into electrical energy for energy storage. Piezoelectric materials are considered the most promising energy harvesters due to their easy preparation, low cost, and light weight. Meanwhile, piezoelectric materials can be used as mechanical sensors to record mechanical deformation based on the frequency and magnitude of the output current. Due to their ability to collect and detect small vibration energy, they can not only detect body movements such as joint movements, but also collect physiological signals such as respiration and heart rate.

Currently, materials with piezoelectric properties discovered include single crystals (such as ammonium dihydrogen phosphate, quartz, etc.), ceramics [57,58], polymers [59], composite materials [60], etc. [20,61,62]. Widely studied piezoelectric ceramic materials, such as zinc oxide, barium titanate, barium zirconate, and potassium sodium niobate, etc., have been extensively studied due to their high voltage electrical coefficients. However, due to their generally high hardness and poor flexibility as thin film or bulk materials, their use in micro and daily wearable devices is limited. At the same time, the emergence of more flexible, stretchable, and lightweight piezoelectric polymers has given them better advantages in the piezoelectric field. In addition, they have advantages such as biocompatibility, chemical stability, and ease of synthesis [63]. At present, the most commonly used piezoelectric polymers include polyvinylidene fluoride (PVDF) [64] and its copolymers such as polyvinylidene fluoride-trifluoroethylene (PVDF-TrFE), polyvinylidene fluoride-hexafluoropropylene (PVDF-HFP), etc. PVDF is a polymorph that exhibits five different crystal forms, each with different chain conformations α (TGTG’), β (TTTT), γ (T3GT3G’), δ (polarized α), and 𝜖 stages [20], as shown in Figure 2a [65]. The most stable phase among them is the nonpolar α-phase, but the phase with the best piezoelectric performance is the β-phase of PVDF, which produces an electric dipole moment due to the asymmetric distribution of negatively charged fluorine ions [66]. Therefore, various strategies have been applied to increase the content of β-phase in PVDF, such as mechanical stretching, strong electric field stretching, annealing, rapid cooling, doping, or multi-material composites. Hu et al. [66] investigated the effects of three typical electrospinning parameters on the composition and crystallinity of β-phase, including applied voltage, needle diameter, and feed flow rate. Research has found that as the applied voltage increases β-phase. Figure 2b shows that as the voltage increases, the phase content and crystallinity first increase and then decrease, and there exists an optimal value. The phase fraction increases parabolic with the increase of feed flow rate, but the crystallinity decreases linearly. Subsequently, in the study of the mixed effects of the three parameters, it was found that the feed flow rate shows the greatest impact on the formation of β-phase. Finally, a new strategy was proposed to improve the performance of electrospun PVDF fibers by prioritizing the electrospinning parameters. Stachewicz et al. [67] proposed that relative humidity shows a significant impact on the crystallinity of β-phase and energy harvesting efficiency of electrospun PVDF, as shown in Figure 2c. The research found that the crystallinity of the β-phase is as high as 74% at a relative humidity of 60%, which increases the d_33_ piezoelectric coefficient of PVDF fibers by more than three times, achieving a power density of 0.6 μW·cm^−2^. More importantly, this study summarized the key factors affecting the piezoelectric performance of PVDF fibers based on humidity variables, which are crystallinity and surface chemistry. Therefore, electrospinning technology is an effective technical means that can use one single step to obtain a large amount of efficient energy conversion PVDF fibers. Lately, Mandal et al. [68] proposed that the δ-phase shows the same excellent piezoelectric performance compared with β-phase, shown in Figure 2d. To confirm the piezoelectricity and ferroelectricity in δ–PVDF nanoparticles, a piezoelectric response force microscopy study was conducted, which exhibited a huge piezoelectric coefficient (d_33_) of approximately −43 pm/V and optimal phase reversal of (Δφ ≈ 180°) at a low applied voltage (i.e., ~±5V).

Zhu et al. [69] combined applied high pressure with a spherical path to form a fiber membrane on a rotating collector to obtain P(VDF-TrFE) NFs with pure β-phase, as shown in Figure 3a. The fiber sensor can output a voltage of up to 18.1V, an output current of 0.177 m µA, and a power density of 22.9 mW cm^−2^. Even under small mechanical stress, it can generate a few volts of voltage, making it a self-powered detector for biomechanical movements of feet, elbows, and fingers. Figure 3b shows that Forouzan et al. [70] used high-speed rotating drums to fabricate pads with arranged nanofibers. Next, the preparation process uses a standard twisting device to twist the ribbon on the x-y plane in the Z direction to produce a single layer of yarn. They proposed that the piezoelectric response of the woven nanogenerator can be enhanced by reducing yarn linear density and increasing twist per meter, the number of ply layers, and fabric density. A detailed study was conducted on the influence of geometric parameters such as yarn linear density, twist per meter, ply layer, weft, and warp density on the piezoelectric performance of PVDF electrospun yarns. By developing twisted yarns of electrospun nanofibers and studying their mechanical and piezoelectric properties, the relative advantages of directional structured yarns over irregular geometric shapes were evaluated. In order to provide electrospun films with a core-shell structure, the coaxial electrospinning method is used to electrospun different films, as shown in Figure 3c. Sun et al. [71] used PVDF, PAN, and coaxial electrospinning techniques to construct a TiO_2_@PVDF/PAN electrospun membrane for applications in photocatalysis and oil-water separation. In the study, the electrospun membrane with PVDF as the sheath and PAN as the core structure shows a large diameter, high strength, and rough surface, which greatly improves its catalytic and oil-water separation performance. Under UV light, the photocatalytic performance can reach 97%. The oil flow rate is 40,163.79 L m^2^ h^−1^, and the oil-water separation rate is as high as 99%. Meanwhile, the membrane maintains a high flow rate after multiple cycles, making it an important candidate for treating industrial and domestic wastewater.

Liu et al. [72] developed a yarn technology for preparing sheathed PVDF/graphene carbon fibers (PVDF/G-CF) through conjugated electrospinning, which uses commercial carbon fibers as the core and electrospun graphene-doped PVDF as the sheath, greatly improving the fatigue resistance and high degree of freedom of PVDF fibers, as shown in Figure 3d. The power density of PVDF/G-CF electronic textiles woven from yarn on a large area reaches 25.5 mW^−2^, and their close distribution gives them excellent softness, washability, and durability. Therefore, it has shown great potential in pressure sensing, self-powered operation, and motion detection. Hassanin et al. [73] introduced the preparation of multifunctional nanofiber membranes using wet electrospinning technology. A hybrid structure of piezoelectric multifunctional sensors was prepared by wet electrospinning PVDF nanofibers onto a poly (3,4-ethylenedioxythiophene)-poly (styrene sulfonate) (PEDOT: PSS) coagulation bath. The schematic diagram of the wet electrospinning process is shown in Figure 3e, with a distance of 15 cm between the high-pressure needle and the PEDOT: PSS collector bath. The electrospun nanofiber membrane prepared by the wet method exhibits a variable surface roughness of up to 120 pm, thereby demonstrating good strain-sensing ability. In addition, the work examined the piezoelectric response and strain-sensing performance of wet-spun nanofibers subjected to over 100 cycles of tensile strain. The proposed work has a high potential for application in wearable flexible devices and mechanical transducer systems.

Finally, different structural designs and assembly methods can significantly improve the electromechanical conversion efficiency of the nanogenerator. Figure 4a shows a typical assembly structure of an energy-harvesting nanogenerator [60]. A simple energy collector is formed by combining a nano composite fiber membrane based on electrospun non-woven PVDF/BaTiO_3_ NW with a conductive electrode film on top and bottom. However, due to the difficulty in exporting the negative charge generated by PVDF, researchers improved the device. Hu et al. [74] used surface energy gradient and push-pull effect to composite heterogeneous PVDF fibers with conductive MXene/CNTs electric spray layers to build an energy collector with hydrophobic-hydrophilic difference, which realized one-way water transfer, thus achieving the effect of self-absorption of sweat Figure 4b. Fan et al. [75] developed an all-nanofiber piezoelectric nanogenerator with a sandwich structure that combines electrospinning and electrostatic direct writing, as shown in Figure 4c. The generator consists of a middle (P(VDF-HFP))/ZnO electrospun nanofiber piezoelectric layer, upper and lower (P(VDF-HFP))/ZnO nanofiber membranes, and a 110 nm Ag layer as the electrode layer. Among them, the added ZnO nanoparticles not only improve the piezoelectric output but also give it over 98% antibacterial function. Due to the fact that the all-nanofiber layer is only 91 µm thickness and does not require further encapsulation, it presents a high breathability of 24.97 mm/s, demonstrating enormous potential in flexible self-powered electronic wearable devices and physical health monitoring. Figure 4d shows an energy harvesting device assembled with MXene coating as the negative frictional layer and PBU fiber as the positive frictional layer [76]. The device includes MXene/fabric and PBU/Ag fabric components connected using copper conductive tape. The working principle of this device is to generate positive charges on the surface of MXene and negative charges on the PBU membrane when external forces interact with MXene and PBU fibers. Due to different electron affinities, electrons will be transferred from MXene to PBU fibers. Due to the opposite charges on the surfaces of the two frictional layers, electrical balance is achieved without generating current or potential. As the two charged layers gradually separate, opposite charges on the surface migrate from the MXene coated conductive fabric electrode to the Ag coated conductive fabric electrode, generating voltage and current.

More recently, Chandran et al. [77] prepared assembled electrospun PVDF nanofibers using paper as the reverse material, which exhibited excellent output electrical performance. The output voltage was 430 V, the short-circuit current density was 0.9 mA/m^2^, and the peak power density was 0.6 W/m^2^. By collecting biomechanical energy from human movement, finger tapping, wrist bending, and elbow bending, the assembled device can power various electronic devices such as calculators and digital thermometers.

### 3.2. Environmental Monitoring

With the acceleration of human industrialization, environmental pollution is increasing rapidly, and pollution detection has become a hot topic in the research field. The nanofibers prepared by the electrospinning method show obvious advantages such as designable micrometer structure, large specific surface area, and simple and inexpensive preparation method, which exhibit important application value in the preparation of chemical sensors for environmental pollution detection. Secondly, flexible microsensors with degradability, biocompatibility, and reusability prepared by electrospinning can monitor environmental changes in real-time to the greatest extent possible. The following is an introduction to electrospun chemical sensors used for monitoring pollutants such as gases, humidity, heavy metals, etc.

#### 3.2.1. Gas Sensors

The Earth is surrounded by the atmosphere, which can absorb a large amount of strong ultraviolet radiation and provide oxygen stability for humans. Oxygen and air environment are important factors for human survival, therefore gas monitoring as an important part of environmental monitoring has received widespread attention and research. In recent years, due to environmental pollution caused by human activities, particulate pollutants, and gas solvents have caused irreparable pollution to the atmosphere, leading to widespread global temperature rise and ecological damage [78]. Among them, small doses of toxic gases generated from industrial and human waste can lead to human poisoning or chronic harm to the body. Therefore, real-time detection of the concentration range of toxic gases can ensure that the human activity area is within a safe range [79]. There is much literature related to the preparation of nanofibers using electrospinning, among which the structural design using this method can significantly improve the sensing performance of gas sensors. In this section, we will demonstrate different structural designs to prepare gas sensor components with different functional performances. 

Hydrogen sulfide (H_2_S) is produced from industrial waste gas, sewage, food processing, etc. It is a colorless, flammable, highly toxic gas with characteristic odors such as rotten eggs [80]. Park et al. [81] utilize porous nano morphology to improve the interaction between the surface of composite materials and sulfate molecules, thereby enhancing the H_2_S gas sensing performance. Researchers used electrospinning to prepare SnO_2_-CuO with a one-dimensional highly porous structure, as shown in Figure 5a. Through the Kirkendall effect, nanofibers were transformed into nanotubes with hollow nanostructures based on the different diffusion rates between SnO_2_-CuO and Sn/Cu. The prepared SnO_2_-CuO nanotubes exhibit significantly enhanced response (R_a_/R_g_ = 1395), with a very fast response time (5.27 s), and a stable recovery time from low concentrations of H_2_S to 5 ppm at 200 °C. Due to the significant increase in specific surface area caused by the porous nano morphology, this gas sensor has lower operating temperature and higher H_2_S sensing performance, making it a promising candidate for gas sensor systems. Nitrogen dioxide (NO_2_), as a representative pollutant gas, can cause serious environmental and health problems, mainly generated by industrial waste gas and automobile exhaust. Due to the fact that the concentration of NO_2_ in environmental and biological exhaled gases typically fluctuates around one billionth (ppb), accurate detection of ultra-low concentrations of NO_2_ is extremely important for achieving pollutant warning [82]. At present, commercial spectroscopic methods such as gas chromatography, mass spectrometry, and spectrophotometer are widely used for NO_2_ analysis. However, these methods often require expensive equipment, fixed operating locations, and high voltage supply. Therefore, achieving real-time detection of harmful gases such as NO_2_ is extremely difficult. In recent years, micro gas sensors assembled with metal oxide semiconductor nanofibers such as indium oxide (In_2_O_3_) nanowires have become the most promising candidate for real-time monitoring of NO_2_ due to their excellent sensing performance and high response sensitivity. Han et al. [83] proposed a sensor based on In_2_O_3_/ZnO yellow shell nanofibers, shown in Figure 5b, which shows significant sensitivity (RG/RA = 6.0) and fast response/recovery time (≤36.68 s) to 1ppm NO_2_ under ultraviolet (UV) irradiation. This performance is significantly improved than that under dark conditions. This is mainly due to the improved photogenerated charge separation efficiency of the heterojunction introduced by the core-shell structure, the reaction sites with increased high specific surface area of the composite structure, and the enhanced NO_2_ receptor function given by gas adsorption. Triethylamine (TEA) is a typical volatile organic compound released from decaying fish and shellfish. Excessive inhalation or contact can cause serious health hazards to the human body, such as irritation to the eyes, skin, and respiratory tract, and even damage to the nervous system. Therefore, there is a great need for high-performance and highly selective TEA gas sensors [84]. Yang et al. [85] prepared a new type of porous NiO/NiFe_2_O_4_ in tube fiber nanostructure by electrospinning method, shown in Figure 5c. By introducing the second component of NiO, the baseline resistance of the NiFe_2_O_4_-based gas sensor was significantly reduced, and the gas sensing performance of the sensor was greatly improved. At the same time, due to its unique fiber nanostructure in the tube and the heterojunction effect between NiFe_2_O_4_ and NiO, the device exhibits good moisture resistance, and relatively small resistance change in a wide humidity range can ensure the stability of response sensitivity. In recent years, chemical resistance gas sensors have attracted extensive attention in the field of harmful gases such as alcohol content detection and environmental quality monitoring due to their advantages of low cost, high stability, good repeatability, and simple process [86]. Wang et al. [87] designed a kind of NiO/In_2_O_3_ parallel two-component heterojunction nanofibers by using the self-made V-type electrospinning technology, as shown in Figure 5d. They not only used the synergistic reinforcement of the two components and their surfaces, but also used the reinforcement of the heterojunction and solid solution on the formation of the depletion layer, and obtained INO-10 SBHNFs with a response sensitivity of one order of magnitude higher than other nanofibers based on In_2_O_3_ at 260 °C. This assembled sensor also presents a high response recovery rate (4 s/36 s) and low detection limit (<200 ppb). In addition, the sensor also shows good selectivity and long-term stability. Hydrogen is an environment-friendly and pollution-free energy, which is a better choice than traditional energy. However, as hydrogen can cause an explosion when its concentration in the air reaches 4.65–93.9 vol%, real-time monitoring of hydrogen in the environment is very important [88]. Chen et al. [89] modified In_2_O_3_ nanofibers prepared by electrospinning with Pd particles to obtain Pd−In_2_O_3_ composite nanofibers, as shown in Figure 5e. The sensor with 1.5 wt% loading showed good selectivity, high response (R_a_/R_g_ = 293.6), short response time (12 s), and recovery time (23 s) to 10,000 ppm H_2_ at room temperature, which was attributed to the catalytic performance and chemical sensitization of Pd, as well as the high specific surface area of porous structure. The composite nanomaterials show excellent sensing properties and are suitable for the detection of low-concentration H_2_ at room temperature. Wang et al. [90] combined the hydrothermal method with electrospinning technology to prepare an H_2_S gas sensor composed of copper oxide nanoflower-modified cobalt oxide nanofiber composite, as shown in Figure 5f. The larger specific surface area of electrospun fibers brings more active centers, which promotes the adsorption of gas on the material surface. Combined with the P-P heterojunction effect on the contact interface between CuO and Co_3_O_4_, the sensor performance is greatly improved. The response sensitivity of the CuO/Co_3_O_4_ composite nanofiber sensor can reach 194% @25 ppm, and the response/recovery time can reach 6 s/25 s@ 25 ppm. At the same time, it has good repeatability, long-term stability, and selectivity. Humidity control is crucial in fields such as industrial manufacturing, healthcare, and air quality, therefore real-time monitoring of humidity is gradually developing towards micro and flexible devices.

#### 3.2.2. Humidity Detector

Humidity control is crucial in fields such as industrial manufacturing, healthcare, and air quality, therefore real-time monitoring of humidity is gradually developing towards micro and flexible devices. In humidity-sensing materials, polymers prepared based on electrospinning have been widely studied and studied due to their sensitive humidity-sensing properties and miniature volume. 

Wang et al. [91] combined electrospinning with chemical vapor deposition technology to prepare a self-powered flexible humidity sensing device based on polyvinyl alcohol/Ti_3_C_2_Tx (PVA/MXene) nanofiber membrane and monolayer molybdenum diselenide (MoSe_2_) composite, as shown in Figure 6a. Integrating composite humidity sensing materials onto a flexible substrate of ethylene glycol phthalate enables them to be worn on different parts of the human body, utilizing the energy generated during movement to provide electrical energy. The peak output can reach 35 mV, and the power density can reach 42 mW m^−2^. Humidity sensors can detect changes in the resistance of polymer nanofiber devices, as shown in Figure 6b. The integrated sensing device can achieve a response sensitivity of 40 with the assistance of a single-layer MoSe_2_, and a response/recovery time of only 0.9/6.3 s. It also has a low hysteresis of 1.8% and good repeatability. Therefore, the self-powered flexible humidity sensor has a good ability to detect human skin humidity and environmental humidity, providing good reference value and development potential for the integration of piezoelectric nanogenerators in the application field of self-powered flexible high-performance humidity sensors. Chen et al. [86] synthesized Fe_2_O_3_@Co_3_O_4_ double shell nanotubes using a simple coaxial electrospinning method, as shown in Figure 6c. By introducing Fe_2_O_3_ to enhance conductivity and adjusting the oxygen vacancy defect content in Co_3_O_4_, the conductivity was significantly improved. The response performance of this composite fiber is 3.5 times higher than that of pure Co_3_O_4_ fibers while exhibiting a lower detection limit (1 ppm). The enhanced gas sensing performance can be attributed to the double-layer hollow structure and heterojunction. The enhanced sensing mechanism of this structure can be explained by the oxygen adsorption model combined with the band structure diagram. Most importantly, they can promote the active sites for gas adsorption on the surface of the sensing layer. At present, self-powered wearable devices that monitor environmental humidity and pollutants in real-time, and have comfortable wearing, sustainable use stability, and electrostatic discharge safety performance have attracted much attention. Sardana et al. [92] used electrospinning to prepare highly ordered MXene nanofibers and cellulose acetate to obtain an anisotropic frictional nanogenerator, which exhibits very high response sensitivity in the rapid adsorption/desorption of water molecules, as shown in Figure 6d. This flexible self-powered sensing device can also easily detect moisture on the skin surface and exhibit good antibacterial performance, making it very suitable for long-term wear.

#### 3.2.3. Other Pollution Detectors

The serious pollution caused by heavy metals to the environment is a serious and urgent problem that affects human health. The sensing materials used for detecting heavy metals have attracted widespread attention and research. Ngoensawat et al. [93] fabricated a conductive nanofiber composite containing poly (3,4-ethylenedioxythiophene) (PEDOT) and silver nanoparticles (AgNPs) prepared by emulsion electrospinning, as shown in Figure 7a. The heavy metal ion sensor assembled on the surface of the screen-printed carbon electrode using this conductive composite material presents a good electrochemical response and can simultaneously detect heavy metal ions, namely Zn (II), Cd (II), and Pb (II), in the mixture by square wave anodic stripping voltammetry. Furthermore, after adding Bi^+3^ to the detection system, the bismuth formed on the electrode can effectively form an alloy with the heavy metal ions deposited after reduction, allowing the detection limits of Zn (II), Cd (II), and Pb (II) to reach 6, 3, and 8 ppb, while maintaining a linear range of 10–80 ppb.

Fotia et al. [94] used electrospinning to composite three different polymers with disodium ethylenediaminetetraacetic acid and ethylenediaminetetraacetic acid as the active layer of the sensor, as shown in Figure 7b. When the active layer reaches equilibrium in the buffer solution of salt dissociation, Na_2_EDTA fibers are encapsulated by porous PS fibers, forming a negative charge distribution on the surface of the active layer barrier. When Pb^+2^ ions are present, EDTA^−2^ can combine with Pb^+2^ ions to form stable complexes, which are proportional to the content of lead ions. Subsequently, as the concentration of pollutants increased, the negative charge of EDTA decreased, hindering the continued generation of PS-Pb/EDTA complexes. The optimal performance of the active layer is achieved by controlling the type of polymer, degree of salt encapsulation, and salt content. Obtained electrospun sensing devices are suitable for environmental monitoring, with a detection range of 10–100,000 μg L^−1^, and a detection limit is 0.031 μ G L^−1^. In addition, it is found that this sensing layer shows good anti-interference ability against interfering pollutants such as thallium (another related heavy metal). Wang et al. [95] prepared the multistage pore structure legume-like UiO-66-NH_2_@carbon nanofiber aerogel-modified electrode through one-step electrospinning. By increasing the coordination number of active site N and HMIs, the enrichment ability of the assembled sensor was improved. Figure 7c shows that heavy metal ions are enriched and reduced to a metallic state on the electrode surface after applying a constant negative voltage. Afterwards, during the stripping stage, the potential of the electrode gradually increases, causing the metal to be oxidized and generating oxidation/stripping peak currents. The UiO-66-NH_2_@carbon nanofiber aerogel modified electrode with multistage pore structure as an electrochemical sensor increases the conductive path and HMIs transmission channel, thus exhibiting ultra-high sensitivity and low detection limit, and can be used for the detection of water quality in the actual environment. This new type of sensor device has excellent performance such as high sensitivity, stability, and anti-interference, thus demonstrating enormous potential in heavy metal ions monitoring. Mercury is undoubtedly a highly toxic water pollutant, and excessive mercury can cause damage to organs such as the brain, lungs, and nervous system of organisms. Therefore, research has been conducted on low-concentration Hg(II) detection. Rotake et al. [96] utilized mercaptosuccinic acid cross-linked pyridinedicarboxylic acid on top of indium-doped ZnO nanofibers, improved charge transfer properties with indium doping, and achieved the detection limit of 3.13 fM by combining MSA-PDCA with multifunctional strategies, resulting in the highest stability constant and strong coordination covalent bonding between carboxyl groups (-COOH) and Hg(II) ions. The sensing parameters are observed through changes in charge transfer resistance, which is proportional to the mass of Hg(II) ions, as shown in the bottom of Figure 7d. The self-assembled portable electrochemical sensor can detect Hg(II) ions in the femtosecond range with ultra-sensitivity.

### 3.3. Pollution Purification

Electrospinning fibers have also attracted attention in pollution control due to their ability to filter pollutants in the air, adsorb heavy metals in water, and catalyze the degradation of organic pollutants in water. At present, researchers have prepared a variety of pollutant filtration materials through different structural and compositional designs. Meanwhile, due to its excellent biocompatibility, degradability, and reusability, it does not cause secondary pollution to the environment, making it the most potential material for pollution control. This section summarizes various materials that have recently been prepared using electrospinning and have shown excellent performance in environmental governance.

#### 3.3.1. Air Filter

Air pollution poses an increasingly serious threat to human health, and the presence of extremely fine particulate matter (PM) and small droplets suspended in the air is a contributing factor and evaluation indicator of air pollution. Among them, particles smaller than 2.5 μm (referred to as PM_2.5_) are considered one of the important pollutants due to their ability to penetrate the lungs and bronchi of the human body, causing unimaginable damage to human health. Therefore, indoor people install air purifiers to obtain a good air environment, and in unavoidable external environments, pollutants can only be filtered by wearing masks. Therefore, the filtration of masks shows important applications in protecting people’s environmental health. Recently, electrospun fiber membranes have been regarded as an important production method for mask filters due to their beneficial mass production advantages such as high efficiency, convenience, and low price.

The ideal structure of an efficient PM_2.5_ filter is a cross-linked mesh structure and secondary structure, which can achieve efficient physical interception of particulate pollutants, as shown in Figure 8a. Modifying its surface with a convex particle structure can increase the polarity adsorption of the network structure, achieving the best effect of air pollutant removal. Deng et al. [97] constructed a graded T-PANa/PVA fiber membrane by combining electrospinning with ultrasonic dispersion and thermal crosslinking techniques to regulate the microstructure and surface chemical groups of fibers, thereby significantly improving the removal rate of PM. The modified hierarchical fiber membrane shows excellent filtration performance, as well as excellent water resistance and good mechanical properties, and is expected to be applied not only in air filters but also in masks for daily filtration. The application of multi-hierarchical electrospun membranes in air filtration masks has attracted widespread attention. However, the nanofibers prepared by traditional electrospinning face the problem of consuming large numbers of organic solvents and causing secondary pollution to the environment. Deng et al. [98] proposed a one-step green electrospinning method to prepare environmentally friendly curved nanobelts membrane, achieving efficient, breathable, and sustainable air filtration, as shown in Figure 8b. The bent ribbon fibers present a loose and elastic structure, significant breathability, and stretchability, and are easier to intercept and polar adsorb pollutants, resulting in excellent removal efficiency for PM. This multi-level structure of environmentally friendly, breathable, and high-performance air filtration membrane provides a new environmentally friendly and green approach for sustainable filtration devices in the future. Masks with bactericidal effects can effectively prevent the transmission of pathogens through the air, providing a new research approach for high-performance masks with public health protection. Currently, most commercial masks can effectively block the spread of particulate matter and airborne bacteria, but their bactericidal effect does not reach an effective level of protection. He et al. [99] constructed a double-layer composite filtration membrane with antibacterial activity by combining Ag/Zn fine particle-modified cotton fabric with electrospun polyvinylidene fluoride/polystyrene nanofibers, as shown in Figure 8c. This structure activates internal tissues through external moisture to generate electrical stimulation, thereby endowing the fabric with antibacterial properties. The fabric can achieve a filtration performance of 99.1% and 79.2Pa for PM_0.3_, and its resistance is demonstrated by a retention rate of 99.64% and 98.75% for *Escherichia coli* and *Staphylococcus aureus* within 20 min, respectively. The efficient disinfection function of this double-layer antibacterial fabric can ensure better bacterial isolation and protection in daily mask applications.

#### 3.3.2. Water Purification

Water is an important factor for human survival, and with the increase in human activities, water pollution is becoming increasingly severe. There are various harmful components in wastewater, so it is necessary to treat different pollutants or prepare multifunctional adsorption and degradation materials that can simultaneously remove multiple pollutants. At present, the removal of different types of pollutants can be divided into two types: heavy metal ion removal and dye/organic pollutant removal [17].

Due to their inability to degrade naturally, heavy metal ions accumulate over time and cause irreversible damage to both humans and ecosystems. Therefore, the removal of heavy metal ions from water pollution is of great significance. Electrospun polymer nanofiber membranes are widely used for removing metal ion materials due to their low cost, good flexibility, easy preparation process, and designable microstructure.

Li et al. [100] utilized the Lewis basicity of nitrogen-containing adsorption sites to regulate the fluorinated groups, better matching with In (III) ion cations, resulting in significant differences in the binding of adsorption sites to them, shown in Figure 9a. In addition, a targeted program is adopted for desorption technology based on the potential energy difference of adsorption sites. Deep separation of In (III) and Fe (III) ions was achieved, and an abnormal concentration of In (III) ions of 20.7 mmol L^−1^ was achieved in an initial desorption solution with a very high mass ratio (m_In_/m_Fe_ = 151.7). This method provides a new technique for the deep separation of In (III) and Fe (III) ions in a mixed system. Yang et al. [101] then used in situ growth to modify the zeolite imidazole salt framework on the surface of electrospun polyacrylonitrile nanofibers, obtaining ZIF-8@ZIF-8 /PAN composite nanofibers, Figure 9b. The obtained nanofibers show a uniform morphology and porous structure, effectively overcoming the crystal aggregation problem of ZIF-8. The abundant active sites brought by its high specific surface area (871.0 m^2^ g^−1^) provide powerful conditions for the effective adsorption of Cr (VI) and reduction of heavy metal ions, with a loading efficiency of up to 82.9%. It can effectively remove Cr (VI) from aqueous media, and partially reduce toxic Cr (VI) ions to relatively harmless Cr (III), with a maximum adsorption capacity (qmax) of 39.68 mg^−1^ and good recovery efficiency. The structure of MOFs loaded and grown on this electrospun nanofiber can also be extended to other water-stable MOF materials, providing the possibility of removing various heavy metal ions. Wu et al. [102] obtained a chitosan (CS) cellulose-based porous nanofiber membrane for the effective removal of heavy metals through the core-shell electrospinning method as shown in Figure 9c. Compared with traditional membranes, core-shell cellulose acetate (CA)/CS bio composite nanofiber membranes have weaker thermal stability, resulting in more copper ions accumulating on the surface. Among them, 30% CS conventional nanofiber membrane showed the best adsorption capacity for copper ions (86.4 mg/g) in an aqueous solution with a pH of 5. The electrospinning technology that utilizes renewable biomass and effective chemical adsorption sites to generate interwoven porous structures has good prospects in water treatment. Xu et al. [103] prepared fluffy SiO_2_-MgO hybrid fiber (SMHF) stacks using a one-step electrospinning method combined with high-pressure steam pretreatment, as shown in Figure 9d, obtaining high specific surface area and multiple surface alkaline sites. When the pH > 4, SMHFs exhibit a distribution coefficient of up to 100 L·g^−1^ for Pb (II) and Cu (II). In terms of performance, the maximum adsorption capacities of SMHFs for Pb (II) and Cu (II) at 298 K are 787.9 and 493.0 mg·g^−1^, respectively. This physicochemical adsorption characteristic can maintain good adsorption capacity and stable repeatability even in the presence of external interfering elements. Due to its efficient adsorption performance, low cost, and simple preparation, this composite fiber has broad potential applications in fields such as environmental protection and governance.

In the world’s environmental problems, water pollution caused by industrial wastewater has attracted high attention. Among them, dyes/organic pollutants are widely and frequently used in various industries due to their important applications. The industrial waste generated is highly toxic and difficult to biodegrade, causing irreversible damage to the entire ecosystem. In order to solve this problem, researchers have prepared various efficient photocatalysts using various organic polymer materials and composite semiconductor materials in recent years. Zhang et al. [104] improved the electrospinning process by utilizing the disintegration of CuBi_2_O_4_ microspheres to form CuBi_2_O_4_ nanoparticles on WO_3_ nanofibers, resulting in a uniformly distributed CuBi_2_O_4_/WO_3_ nanoheterojunction and the assembly of 0D/1D stepped heterojunction nanofiber membrane photocatalysts, as shown in Figure 10a. The expected structure follows the S-type charge transfer mechanism, effectively promoting charge transfer at the interface and effectively solving the high recombination problem of photo-generated carriers in the narrow bandgap semiconductor CuBi_2_O_4_. In addition, this heterojunction promotes the spatial separation of photo-induced electrons and holes, improving the redox ability of the composite material. Excellent photocatalytic activity of antibiotics in environmental remediation and significant improvement in photocatalytic degradation activity of tetracycline under weak visible light irradiation (5 W LED lamp) were achieved, which were 8.1 times and 3.6 times higher than pure WO_3_ nanofibers and pure CuBi_2_O_4_ microspheres, respectively. The 0D/1D CuBi_2_O_4_/WO_3_ Photocatalytic fiber materials, as a classic improvement method for preparing new nanocomposite photocatalysts using electrospinning technology, are also applied in various composite materials [105,106]. As an environmentally friendly and green material, photocatalytic materials have the characteristic of being reusable, but powder photocatalysts are limited in practical applications due to their low recovery rate. Wang et al. [107] used electrospinning combined with coating technology to obtain a new three-dimensional layered photocatalytic fiber membrane consisting of polyacrylonitrile (PAN), polydopamine (PDA), and Tb doped graphitized carbon nitride/ZnIn_2_S_4_ (Tb-g-_3_N_4_/ZnIn_2_S_4_) from bottom to top (PPTZ), as shown in Figure 10b. The multi-layer photocatalytic fiber PPTZ film significantly suppresses the recombination of photo-generated carriers due to the heterojunction formed between the multi-layer semiconductor fibers, resulting in excellent photocatalytic performance under simulated sunlight compared to any single-layer film, with degradation rates 2.1 times and 2.5 times higher than other single-layer samples, respectively. Meanwhile, the PDA layer can serve as an electron transfer medium, effectively transferring electrons from Tb-g-C_3_N_4_ to ZnIn_2_S_4_ and PDA. In addition, this material is non-toxic and pollution-free, and will not cause secondary pollution to the environment in practical applications. Regarding the photocatalytic promotion mechanism of heterojunctions, the most recent research indicates that it can be explained through band theory. Taking g-C_3_N_4_/MoS_2_ composite nanofiber photocatalyst as an example [108], under visible light irradiation, g-C_3_N_4_/ MoS_2_ on PAN electrospun film is simultaneously excited, generating photo-induced electrons and holes. It is known that the conduction band of g-C_3_N_4_ is higher than that of MoS_2_, and the valence band of MoS_2_ is lower than that of g-C_3_N_4_, as shown in Figure 10c. The photoelectrons generated in g-C_3_N_4_ transfer from its conduction band to the conduction band of MoS_2_, while the photo-induced holes generated in MoS_2_ transfer from the valence band to the g-C_3_N_4_ valence band, causing a large number of photoelectrons to gather in the conduction band of MoS_2_. A large number of holes aggregate in the g-C_3_N_4_ valence band, promoting the effective separation of photo induced electrons and holes, significantly reducing the photo induced electron hole recombination rate, thereby improving the photocatalytic efficiency. Finally, Figure 10d vividly depicts the application of photocatalytic materials in water pollution control and environmental protection [109]. It presents that Fe^3+^doping significantly reduces the bandgap of TiO_2_ fibers, thereby shifting the absorption edge of the visible light region of the solar spectrum. Research has found that holes, •O^−2^, and •OH make significant contributions to photocatalytic degradation. The sturdy fiber structure allows the material to maintain stable performance during repeated use. Therefore, electrospinning provides an eco-friendly and practical solution for industrial wastewater treatment due to its advantages of a simple preparation process, low cost, and no secondary pollution, which are suitable for large-scale production.

## 4. Conclusions and Perspectives

The main breakthroughs in electrospinning and its significance in energy and environmental detection and governance were discussed based on recent research findings. Various synthesis and preparation processes were compared, and the effects of different devices and electrospinning conditions on the performance of functional materials were summarized and discussed. The current advanced assembly technology and functional applications in different sub-fields were summarized. Their significant impact in the preparation of highly stable and ultra-sensitive responses in the field of energy and environment were demonstrated. In addition, the contribution of factors such as preparation parameters, environmental factors, combination of other methods, and surface modification of electrospinning significantly enhances the performance of prepared composite nanofibers. Subsequently, electrospun nanofibers have broad prospects in environmental sensing, environmental pollutant quality, air pollution particle filtration, etc. This article summarizes the applications of electrospun polymer nanofibers, including human motion monitoring and energy harvesting, environmental monitoring, and pollution purification.

Compared with traditional piezoelectric materials, hybrid piezoelectric materials can not only generate energy but also collect energy and use it for real-time monitoring, thereby achieving effective utilization and conservation of energy in the environment. Flexible wearable devices assembled with such materials can simultaneously obtain energy from various sources, perceive various fluctuations in the surrounding environment, and transmit wireless data, achieving portable, multi-purpose, and sustainable applications. As described in Section 3.1, different phase contents, structural designs, and assembly methods were summarized to improve the performance of high-voltage electric nanofibers. By adjusting the content of different effective piezoelectric phases, piezoelectric performance can be improved. However, the mechanism of action is not yet clear, and further research is needed to determine more targeted improvements. Although different structural designs and assembly methods can adjust and optimize performance, the relationship between structure and performance is still unclear and requires further exploration and research.

In the field of environmental sensing, compared with other methods such as hydrothermal and template methods, electrospinning, due to its more flexible preparation method, can achieve various special nanostructure designs, such as the porous, hollow, trap, surface nanoflower modification structures summarized in this article. These different structures can increase specific surface area, enhance conductivity, promote charge transfer, and achieve more accurate and rapid detection in complex environments. In addition, due to the inexpensive and easy-to-operate preparation process of electrospinning technology, this method is expected to achieve large-scale production and commercialization, making it a highly promising technological means for practical application in the future.

In terms of environmental pollution purification, this article mainly reviews the application of electrospinning in the fields of air filter and water purification, including structural design, material assembly, purification principles, and its impact on the environment. Further discussion and research are needed to improve the application of composite nanofibers in environmental pollution filtration, including structural improvement, mechanical performance design, and efficiency and stability enhancement. In summary, it is found that porous materials are an ideal structural design for pollution filtration, but porous structures also reduce the mechanical properties of fibers. How to balance the relationship between the two is worth further research? For photocatalytic materials, the catalytic mechanism and dynamic reaction process are not yet clear, and more experimental research is needed to further improve the pollutant decomposition efficiency and cycle stability of photocatalytic materials in a targeted manner. With the continuous in-depth research on electrospinning and composite polymer materials, it is believed that this method can be used in the near future to solve various environmental problems and achieve the governance of environmental and energy issues.

## Figures and Tables

**Figure 1 polymers-16-00839-f001:**
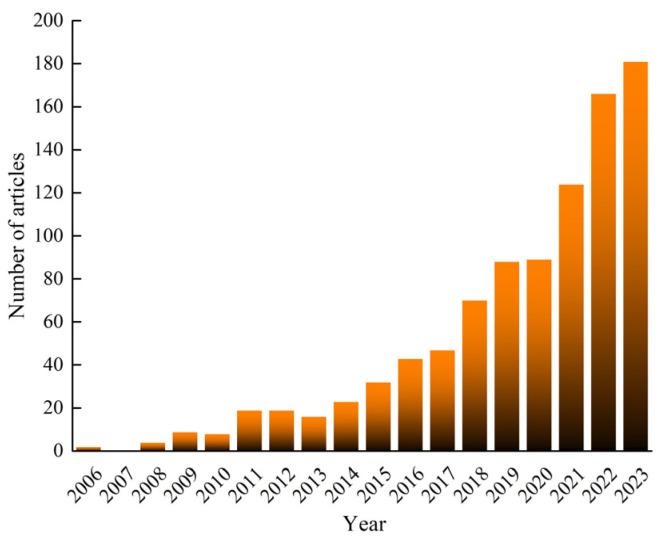
Research statistical data on the theme of “Electrospinning and Energy and Environment” retrieved from the literature on the “Web of Science” platform.

**Figure 2 polymers-16-00839-f002:**
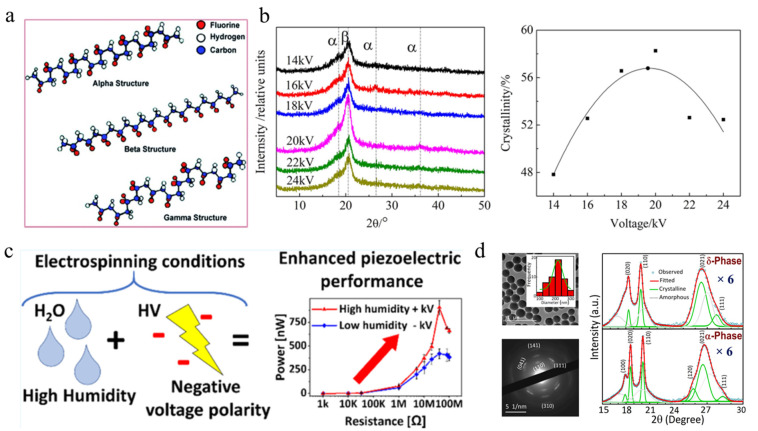
The intrinsic characteristics and influencing factors of electrospun PVDF. (**a**) The schematic diagram for typical α, β, and γ crystalline phases of PVDF [65]. Copyright 2021, reproduced with permission from MDPI. (**b**) The influence of voltage on the XRD patterns and crystallinity of electrospun PVDF nanofibers [66]. Copyright 2021, reproduced with permission from MDPI. (**c**) Schematic diagram of the enhancement mechanism of humidity and voltage polarity on piezoelectric performance [67]. Copyright 2020, reproduced with permission from ACS Publications. (**d**) TEM images δ–PVDF nanoparticles and the difference between the XRD characteristic spectra of phase δ and phase α [68]. Copyright 2022, reproduced with permission from Elsevier.

**Figure 3 polymers-16-00839-f003:**
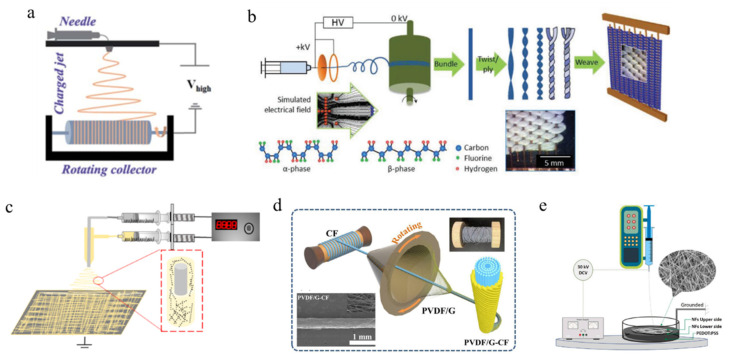
Schematic diagram of different electrospinning collection devices for preparing piezoelectric nanofibers with different structures. (**a**) A classic electrospinning device for preparing P (VDF-TrFE) nanofibers [69]. Copyright 2021, reproduced with permission from RSC. (**b**) Schematic diagram of the fabrication process for copolymer yarns with a geometric arrangement of piezoelectric nanofabrics [70]. Copyright 2020, reproduced with permission from Wiley. (**c**) Preparation of core-sheath structure using coaxial electrospinning technology [71]. Copyright 2019, reproduced with permission from John Wiley and Sons. (**d**) Schematic diagram of the preparation device for conjugated electrospun yarn [72]. Copyright 2023, reproduced with permission from Springer Nature. (**e**) Schematic diagram of preparing piezoelectric nanofiber sensing mat using wet spinning process [73]. Copyright 2023, reproduced with permission from Elsevier.

**Figure 4 polymers-16-00839-f004:**
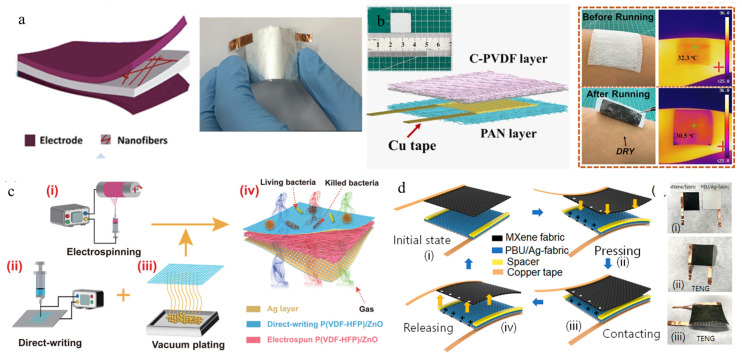
Schematic diagram of different assembly methods for piezoelectric pressure sensors. (**a**) Schematic diagram and actual photo of the assembly structure of a typical piezoelectric pressure sensor [60]. Copyright 2018, reproduced with permission from RSC. (**b**) Schematic diagram of pressure sensor structures with different sensor layer combinations and their optical and infrared images applied on the skin surface [74]. Copyright 2023, reproduced with permission from Springer. (**c**) Assembly of ultra-thin piezoelectric nanogenerators with three-dimensional structure by combining electrospinning (i), direct writing (ii), vacuum plating (iii) and 3D structure of ANF-PENG (iv) [75]. Copyright 2023, reproduced with permission from Wiley. (**d**) A schematic diagram of a self-powered frictional nanogenerator constructed using MXene-coated fabric and PBU fibers deposited on Ag-coated conductive fabric, as well as a physical image of the device [76]. Copyright 2023, reproduced with permission from ACS Publication.

**Figure 5 polymers-16-00839-f005:**
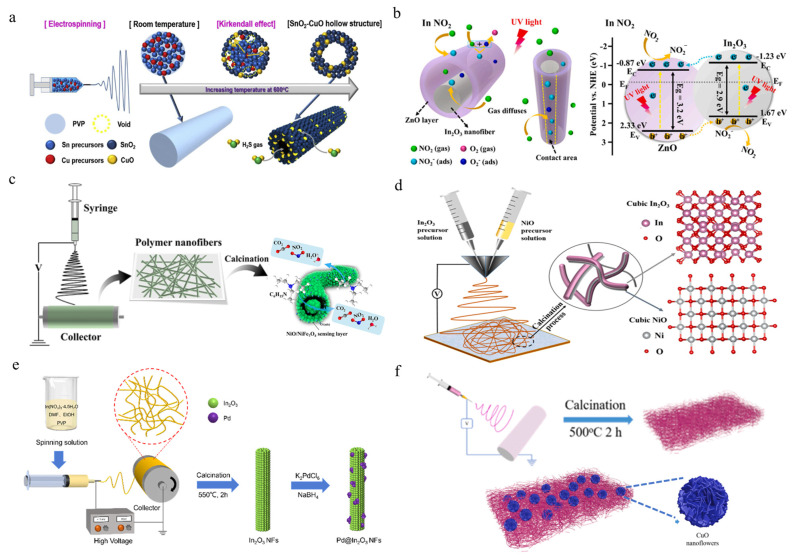
Schematic diagram of the fabrication process for gas sensors with different micro/nanostructures. (**a**) Porous hollow nanofibers prepared by single needle electrospinning with Kirkendall effect to improve H_2_S gas sensing performance [81]. Copyright 2020, reproduced with permission from Elsevier. (**b**) Core-shell nanofibers and heterojunctions for NO_2_ detection, along with corresponding band diagrams [83]. Copyright 2021, reproduced with permission from Elsevier. (**c**) Porous fiber-in-tube nanocomposites for fast triethylamine detection [85]. Copyright 2022, reproduced with permission from ACS Publication. (**d**) Parallel design of bi-component heterojunction nanofibers for high-performance ethanol gas sensors by side-by-side electrospinning [87]. Copyright 2022, reproduced with permission from Elsevier. (**e**) Hydrogen sensors assembled by surface-modified particles on nanofibers [89]. Copyright 2022, reproduced with permission from ACS Publications. (**f**) Electrospun nanofibers modified nanoflowers for rapid response to hydrogen sulfide gas sensors [90]. Copyright 2023, reproduced with permission from Elsevier.

**Figure 6 polymers-16-00839-f006:**
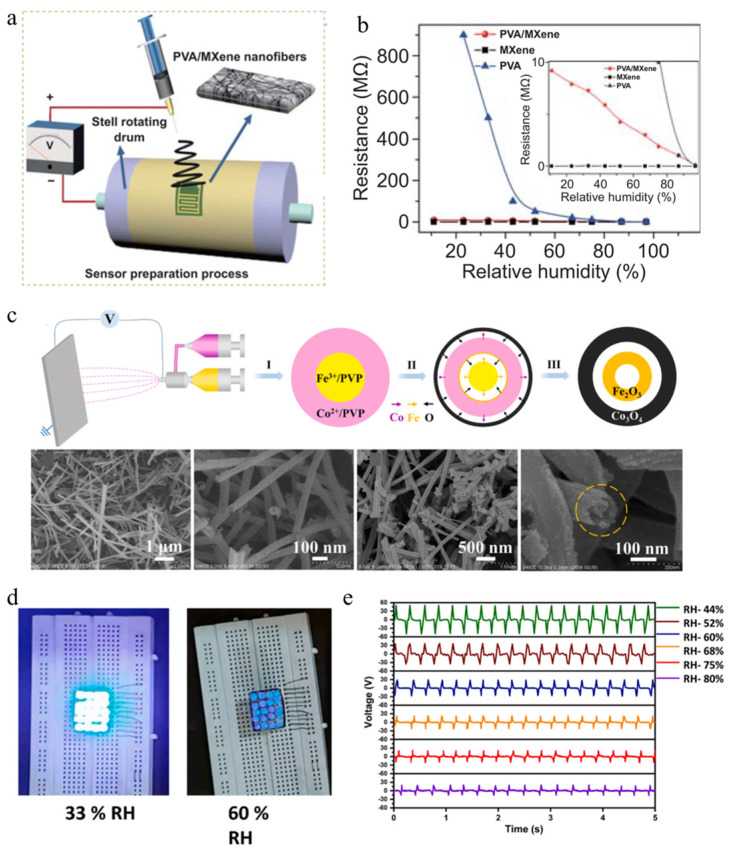
(**a**) Schematic diagram of PVA/MXene composite fiber humidity sensor prepared by electrospinning [91]. Copyright 2021, reproduced with permission from Springer. (**b**) Resistance variation curves of MXene, PVA, and PVA/MXene composite fiber sensors exposed to various relative humidity conditions [91]. Copyright 2021, reproduced with permission from Springer. (**c**) Schematic diagram of electrospinning preparation process and formation mechanism of double-shelled nanotubes, which includes double-shelled structures formed by metal ions and oxygen diffusion during sintering in air, as well as the morphologies of hollow nanotubes and double-shelled nanotubes [86]. Step I, II, III represented the electrospinning process of the precursor nanofiber composites, the metal ions, and oxygen diffusion process during sintering in air, respectively. Copyright 2022, reproduced with permission from Elsevier. (**d**) A photo demonstrating the LED state of a humidity sensor assembled using an electrospun anisotropic frictional nanogenerator at different relative humidity levels [92]. Copyright 2022, reproduced with permission from Elsevier. (**e**) The output voltage of the sensor varies with humidity [92]. Copyright 2022, reproduced with permission from Elsevier.

**Figure 7 polymers-16-00839-f007:**
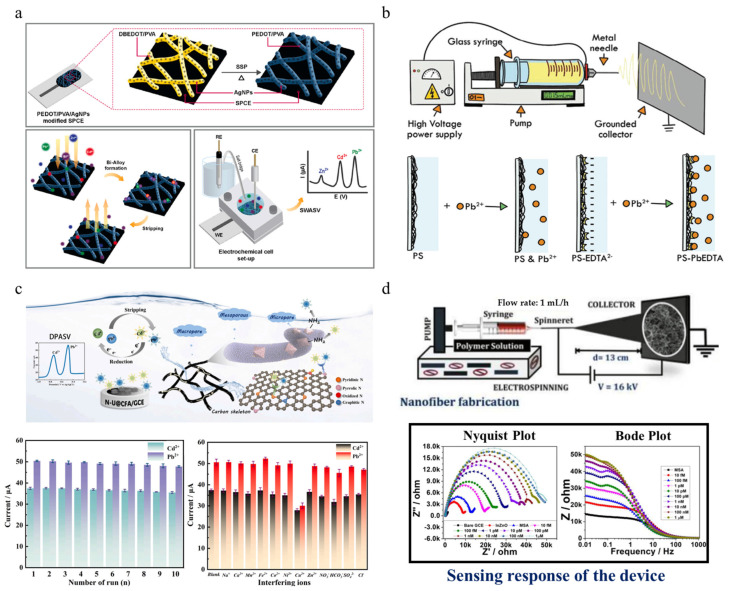
(**a**) Schematic diagram of a heavy metal ion sensor prepared from a screen-printed carbon electrode modified with electrospun PEDOT/PVA/AgNPs fibers, a surface metal bismuth alloy formed by stripping, and the detection equipment composed of an electrochemical battery system [93]. Copyright 2022, reproduced with permission from Elsevier. (**b**) Schematic diagram of the interaction between the active layer surface and ions during the sensing process of Pb ions using polymer fibers prepared by electrospinning [94]. Copyright 2023, reproduced with permission from Wiley. (**c**) The electrochemical detection mechanism of multistage porous legume-like nanofiber aerogel, the repeatability measurement of a small amount of Cd^2+^ and Pb^2+^, and the anti-interference performance of the sensor in a mixed solution containing 50-time interfering ions [95]. Copyright 2023, reproduced with permission from Elsevier. (**d**) Schematic diagram of a portable electrospun-InZnO nanofibers electrochemical sensor platform for selective capture of Hg(II) ions and electrochemical impedance spectroscopy of Hg(II) ions [96]. Copyright 2022, reproduced with permission from Elsevier.

**Figure 8 polymers-16-00839-f008:**
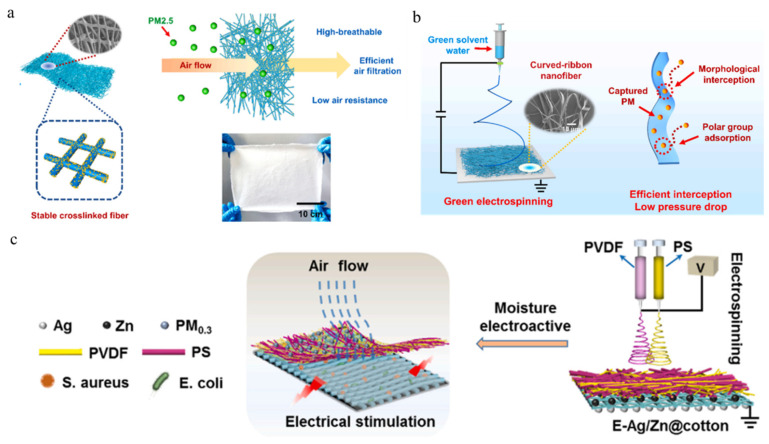
(**a**) Schematic diagram of crosslinked T-PANa/PVA fibers with excellent filtration performance, water resistance, mechanical properties, and optical image of the composite fiber membrane [97]. Copyright 2023, reproduced with permission from Elsevier. (**b**) The curved-ribbon electrospun nanofibers with environmentally friendly breathability and high-performance air filtration [98]. Copyright 2022, reproduced with permission from Elsevier. (**c**) Schematic diagram of double-layer structure composite filter medium [99]. Copyright 2023, reproduced with permission from Elsevier.

**Figure 9 polymers-16-00839-f009:**
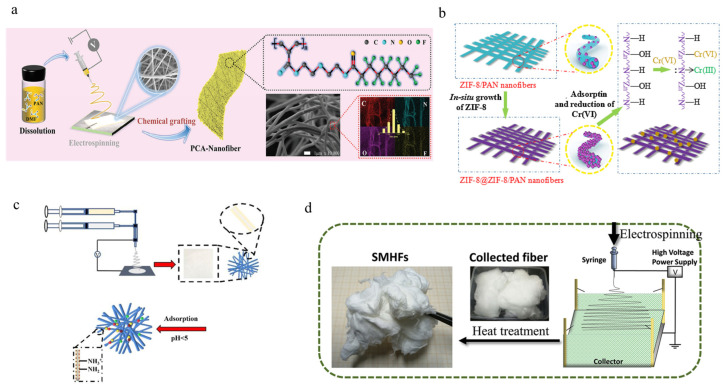
(**a**) Schematic diagram of the preparation process of perfluorooctanoic acid nanofibers, along with SEM and EDS elemental map images and corresponding diameter distribution [100]. Copyright 2023, reproduced with permission from Wiley. (**b**) Schematic diagram of the preparation of ZIF-8@ZIF-8/polyacrylonitrile nanofibers [101]. Copyright 2020, reproduced with permission from Elsevier. (**c**) Preparation and adsorption mechanism of core-shell cellulose acetate biocomposite nanofiber membrane [102]. Copyright 2022, reproduced with permission from Springer. (**d**) Schematic diagram of SiO_2_ MgO mixed fibers prepared by solution electrospinning process [103]. Copyright 2020, reproduced with permission from Elsevier.

**Figure 10 polymers-16-00839-f010:**
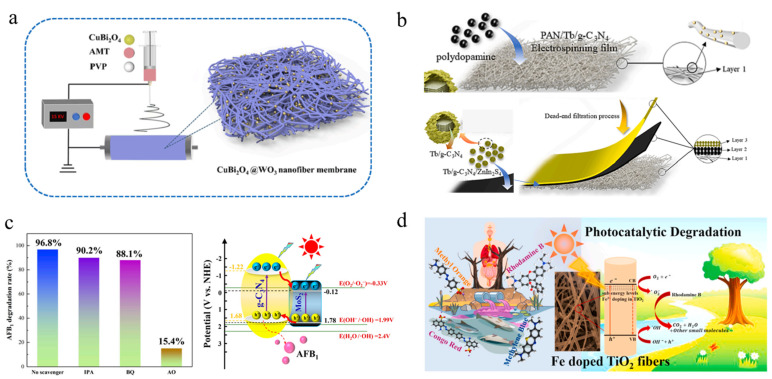
Schematic diagram of electrospinning material design for dye/organic pollutant removal, the classic catalytic mechanism of composite materials, and application of catalytic degradation in environmental protection. (**a**) Typical 0D/1D composite nanofiber membrane photocatalysts utilizing electrospinning method [104]. Copyright 2023, reproduced with permission from Elsevier. (**b**) Schematic diagram of the composition of multi-layer photocatalytic films [107]. Copyright 2023, reproduced with permission from Elsevier. (**c**) Schematic diagram of a photocatalytic mechanism for degradation of the polymer and two-dimensional composite electrospun films [108]. Copyright 2023, reproduced with permission from MDPI. (**d**) Application of green photocatalytic materials in the treatment of industrial wastewater pollution and environmental protection [109]. Copyright 2023, reproduced with permission from Elsevier.

## Data Availability

No new data were created or analyzed in this study. Data sharing is not applicable to this article.

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
