# Peer review of "Advanced Electrospinning Technology Applied to Polymer-Based Sensors in Energy and Environmental Applications"

_polymers, 2024, doi:10.3390/polym16060839_

Round 1

Reviewer 1 Report

Comments and Suggestions for Authors

This review paper provides a thorough overview of current research surrounding electrospinning techniques for energy and environmental purposes. The writing is clear, and the structure is logical. Here's some feedback to further refine the work:

 ·        Abstract: It's mentioned that unresolved issues in detection and environmental applications are identified. Could you please point out the specific sections of the manuscript where these issues are discussed in detail?

·        Section Titles: Sections 3 and 4 both appear titled "Applications in Energy and Environmental Applications." If the fourth section is intended as the conclusion, revising the heading to reflect that would improve clarity.

·        Similarity: The similarity or percentage match is high?

Author Response

1. Abstract: It's mentioned that unresolved issues in detection and environmental applications are identified. Could you please point out the specific sections of the manuscript where these issues are discussed in detail?

Response:

Thank you for the comment. The unresolved issues in the detection and treatment of environmental issues are discussed in detail on page 20, paragraph 3, “However, the mechanism of action is not yet clear, and further research is needed to determine more targeted improvements. Although different structural designs and assembly methods can adjust and optimize performance, the relationship between structure and performance is still unclear and requires further exploration and research.”, and last paragraph “Further discussion and research are needed to improve the application of composite nanofibers in environmental pollution filtration, including structural improvement, mechanical performance design, and efficiency and stability enhancement. In summary, it is found that porous materials are an ideal structural design for pollution filtration, but porous structures also reduce the mechanical properties of fibers. How to balance the relationship between the two is worth further research. For photocatalytic materials, the catalytic mechanism and dynamic reaction process are not yet clear, and more experimental research is needed to further improve the pollutant decomposition efficiency and cycle stability of photocatalytic materials in a targeted manner. With the continuous in-depth research on electrospinning and composite polymer materials, it is believed that this method can be used in the near future to solve various environmental problems and achieve the governance of environmental and energy issues.”

2. Section Titles: Sections 3 and 4 both appear titled "Applications in Energy and Environmental Applications." If the fourth section is intended as the conclusion, revising the heading to reflect that would improve clarity.

Response:

We apology for the unclear title and changed the title of Section 4 from “Applications in Energy and Environmental Applications” to “Conclusions and perspectives”. The revised parts are marked by red in the manuscript.

3. Similarity: The similarity or percentage match is high?

Response:

We have checked and revised the similar parts, for example on page 2, paragraph 2 “Many flexible and wearable electronic devices have been used to detect health-related signals, such as pressure/strain sensors, temperature sensors, humidity sensors, gas sensors, integrated sensing platforms, and etc.”; Page 2, last paragraph, “Electrospinning utilizes high-voltage electrostatic field forces to form Taylor cones from polymer solutions or melts, which are then stretched into countless nanoscale ultrafine fibers, and then cured through solvent evaporation or melt solidification [23–25]. By electrospinning, not only can a morphology controllable nanofiber membrane be obtained, but the membrane also has a relatively large specific surface area and high porosity.”; Page 8, first paragraph, “The oil flow rate is 40 163.79 L m2 h-1, and the oil-water separation rate is as high as 99%. Meanwhile, the membrane maintains a high flow rate after multiple cycles, making it an important candidate for treating industrial and domestic wastewater.”; Page 8, last paragraph, “The proposed work has high potential for application in wearable flexible devices and mechanical transducer systems.”; on Page 17, first paragraph, “The modified hierarchical fiber membrane shows excellent filtration performance, as well as excellent water resistance and good mechanical properties,”, and etc. All revised parts are marked by red in the manuscript.

Reviewer 2 Report

Comments and Suggestions for Authors

This review paper is novel and hence can be recommended after a few minor modifications. These are:

1. Fig. 1 should indicate the increasing number of publications based on electrospun nanofibers.

2. Please incorporate a section indicating differnt types of nanofibers used in literature. 

Author Response

  1. Fig. 1 should indicate the increasing number of publications based on electrospun nanofibers.

Response:

Thank you for the comment. We have added Figure 1 and summarized the number of articles published on electrospun fibers in the fields of energy and environment in recent years (from 2006 to 2023) and added relevant explanations on page 5, first paragraph. “The data in Figure 1 shows that the application of electrospinning in the fields of energy and environment has been increasing rapidly year by year, and has been growing rapidly since 2020. Therefore, this section summarizes the latest progress in various applications of electrospinning in these fields.”

  1. Please incorporate a section indicating different types of nanofibers used in literature.

Response:

Thank you for the comment. We have added “Abbreviations” at the end of the revised manuscript and highlighted it in red. “Cellulose acetate (CA); Carbon nanotubes (CNTs); Hydrogen sulfide (H2S); Indium oxide (In2O3); Metal organic frameworks (MOFs); Molybdenum diselenide (MoSe2); Nitrogen dioxide (NO2); Poly (3-aminobenzylamine) (PABA); Polyacrylonitrile (PAN); Polydopamine (PDA); Poly (3,4-ethylenedioxythiophene) (PEDOT); Polyvinyl alcohol/Ti3C2Tx (PVA/MXene); Polyvinylidene fluoride (PVDF); Poly (3,4-ethylenedioxythiophene)-poly (styrene sulfonate) (PEDOT: PSS); Polyvinylidene fluoride-trifluoroethylene (PVDF-TrFE); Polyvinylidene fluoride-hexafluoropropylene (PVDF-HFP); PVDF/graphene carbon fibers (PVDF/G-CF); Silver nanoparticles (AgNPs); SiO2-MgO hybrid fiber (SMHF); Tb doped graphitized carbon nitride/ZnIn2S4 (Tb-g-3N4/ZnIn2S4); Triethylamine (TEA); Tetraethyl orthosilicate (TEOS); Zeolite imidazole salt framework (ZIF)”.

Round 2

Reviewer 1 Report

Comments and Suggestions for Authors

The revisions made to this paper are satisfactory. I recommend it for publication in its current form.